# Association of APOE e2 genotype with Alzheimer's and non-Alzheimer's neurodegenerative pathologies

Terry E. Goldberg[1✉], Edward D. Huey[2] & D. P. Devanand[2]

The apolipoprotein E (APOE) gene contains both the major common risk variant for late onset Alzheimer's disease (AD), e4, and the major neuroprotective variant, e2. Here we examine the association of APOE e2 with multiple neurodegenerative pathologies, leveraging the NACC v. 10 database of 1557 brains that included 130 e2 carriers and 679 e4 carriers in order to examine potential neuroprotective effects. For AD-related pathologies of amyloid plaques and Braak stage, e2 had large and highly significant protective effects contrasted with e3/e3 and e4 carriers with odds ratios of about 0.50 for e3 contrasts and 0.10 for e4 contrasts. When we separately examined e2/e4 carriers, risk for AD pathologies was similar to that of e4 carriers, not e2 carriers. For multiple fronto-temporal lobar pathologies and tauopathies, e2 was not significantly associated with pathology. In sum, we found that e2 was associated with large but circumscribed protective effects.

---

[1] Psychiatry and Anesthesiology, Columbia University Irving Medical Center, 1051 Riverside Drive, Unit 126, New York, NY 10032, USA. [2] Psychiatry and Neurology, Columbia University Irving Medical Center, 1051 Riverside Drive, Unit 126, New York, NY 10032, USA. ✉email: teg2117@cumc.columbia.edu

Apolipoprotein E (ApoE) is the major lipid transporter in brain. Its encoding gene, APOE, is tri-allelic at two loci in exon 4 that give rise to three haplotypes and six genotypes: e2 has cysteines at aa residues 112 and aa 158; e4 has arginines at these aa sites, and e3 has a cysteine at 112 and an arginine at 158. These substitutions provide the apparent basis for the differing molecular properties among the isoforms, including binding to the low-density lipoprotein receptor (the e2 isoform binds with very low affinity), protein abundance (the e2 isoform is most abundant in brain), cleavage propensity (the e4 isoform is most likely to undergo enzymatic cleavage), amyloid Beta protein (Aβ) interactions, inflammation, and lipidation[1–5]. The e4 variant is the major risk variant for late onset AD with a mean OR = 3.6 when referenced to the "neutral" e3 variant[6]. In contrast, the e2 variant is the major common protective variant for late onset Alzheimer's disease with an OR = 0.54 when also referenced to e3[6]. In human CSF obtained from an Alzheimer's Disease Neuroimaging Initiative sample of healthy older controls, we reported that e2 was associated with increased Aβ levels and reduced p-tau levels when contrasted with e3 homozygotes[7]. These associations suggest an e2 anti-AD profile. Morris et al.[8] observed a similar pattern in an an academic memory clinic samples. In transgenic mouse models of AD with targeted human APOE replacement e2 was associated with reduced levels of amyloid in rodent brain, while e4 promoted amyloid deposition[9].

These studies have focused on clinical diagnoses or rodent models, without examination of neuropathology in humans. This gap in knowledge is important because clinical diagnoses have error rates of 15–30% relative to neuropathological diagnoses that represent the current gold standard for diagnosis[10,11]. No study has examined the association of e2 genotype to the main AD pathologies of amyloid and tau, and other dementia-related pathologies. The need to evaluate these associations is highlighted by recent reports showing that the majority of elderly individuals with dementia have multiple underlying etiologies[12]. Several neurodegenerative disorders share certain features, including protein misfolding and aggregation and perhaps prion-like propagation[13,14]. Insofar as e4 promotes protein aggregation in AD and e2 may reduce it, we examined the impact of these APOE alleles on AD and other proteinopathies including Lewy body dementia involving alpha-synuclein, frontotemporal lobar degeneration (FTLD) related 3R and 4R tau inclusion bodies, TDP-43 cytosolic inclusion bodies, and argyrophilic grain disease.

To implement this strategy, we interrogated the NACC postmortem human brain v10 database because it used up to date pathological criteria, advanced immunohistochemical techniques, and had a large number of cases (>1500). Critically, we elected not to utilize clinical diagnoses for AD or other dementias for our analytic strategy because the large number of sites increases variability in clinical diagnoses and inter-site reliability of diagnoses, including neuropathological diagnoses conducted independently at each site, has not been established in the NACC consortium. Our rationale was to make as few assumptions as possible about clinical phenotypic validity or reliability or even formal diagnostic neuropathologic criteria. Rather, we sought to understand the impact of APOE genotype on amyloid plaques and NFTs, other protein aggregation abnormalities involved in FTLD and related tauopathies (TDP-43, Pick's, PSP, CBD, argyrophilic grain disease), and alpha-synuclein Lewy bodies. We believe that this approach is unbiased and powerful and allowed for the inclusion of all cases in the collection.

## Results

### AD histopathology.
For diffuse amyloid plaques ~35% of e2 carriers were in stage 0 (of 6 levels) indicating absent neocortical plaque involvement. About 65% of e4 carriers were in the most severe stage (5). The Chi square was significant ($X^2 = 263.22$, df = 15, $p < 0.0001$). See Fig. 1a for the relative proportion of APOE genotypes at each severity stage. Planned contrasts of e2 and e3 using ordinal regression were significant as in Table 1, thus, e2 was associated with a 57% reduction in the odds ratio (OR) meeting the criteria for any given stage compared to e3, and an 89% reduction of the OR of meeting the criteria for any given stage compared to e4, i.e., e2 was protective (see Supplementary Tables 1–3 for raw frequency counts). Note that these models (as well as models for Braak stage and neuritic plaques were adjusted for age at death and sex. Sex and age point estimates are in Supplementary Table 4.

APOE genotype was significantly associated with Braak stage by Chi square ($X^2 = 234.67$, df = 18, $p < 0.0001$). (See Supplementary Table 2 for raw frequency counts.) Approximately 40% of e2 carriers, 31% of e3/e3 carriers and 11% of e4 carriers were in Braak stages 0, 1, or 2, indicating no tau pathology or tau pathology restricted to transentorhinal cortex as shown in Fig. 1b. A planned contrast of e2 and e3 using ordinal regression was significant (Table 1). Thus, e2 was associated with a 46% reduction in the OR of meeting the criteria for any given stage when compared to e3, i.e., e2 was protective. Similarly and to an even larger degree, a planned contrast of e2 and e4 using ordinal regression demonstrated that the e2 group was associated with an 88% reduction in the OR for the probability of meeting the criteria for any given severity stage compared to the e4 group. ORs are in Table 1. Sex and age point estimates are in Supplementary Table 4.

For neuritic plaques ~45% of e2 carriers were in stage 0, indicating no neuritic plaques. Only 8% of e4 carriers lacked this pathology. The severity frequency by genotype Chi square was significant ($X^2 = 209.18$, df = 9, $p < 0.0001$). See Fig. 1c for the relative proportion of APOE genotypes at each severity stage. The planned contrast of e2 and e3 using ordinal regression was significant as in Table 1. E2 was associated with a 45% reduction in the OR of meeting the criteria for any given stage compared to e3, i.e., e2 was protective. Similarly and to an even greater degree, the planned contrast of e2 and e4 using ordinal regression was significant as in Table 1. E2 was associated with an 86% reduction in the OR of meeting the criteria for any given stage compared to e4, i.e., e2 was protective. ORs are in Table 1. Sex and age point estimates are in Supplementary Table 4. (As can be inferred from the results above, in all cases e4 carriers demonstrated increased risk in contrast to e3/e3 carriers; ordinal regression data not shown).

### Mediation.
We conducted a mediation analysis in order to determine if APOE e2 contrasted with e3 had direct effects, indirect effects through amyloid, or both types of effects on Braak stage. All paths were adjusted for age at death and sex. The mediation effect through amyloid neuritic plaque extent was highly significant (Sobel statistic = 3.41, $p = 0.0004$) indicating that a mediation effect was present in which e2 influenced tau pathology through amyloid plaque burden. The indirect path coefficient effect was 0.09. The direct path coefficient, in which e2 also had a direct effect on tau Braak stage after mediator adjustment (i.e., the c′ path), was 0.04. Thus, the percent indirect effect was 69 and the percent direct effect was 31. This is displayed in Fig. 2.

### E2/e4 genotype.
We next examined the e2/e4 genotype and its relationship to ABC pathological stages as contrasted to the e2/e3 genotype and the e3/e4 genotype (thus controlling for copy number of the protective and risk variants), as well as the neutral

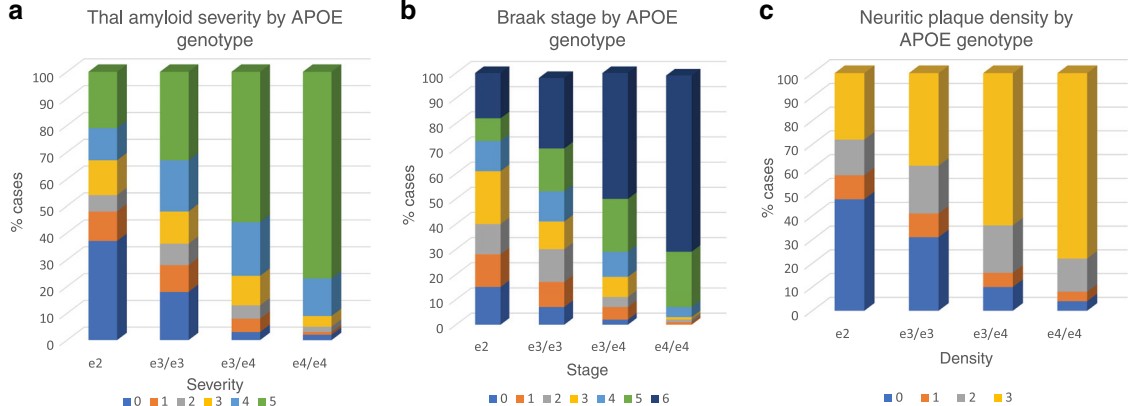

**Fig. 1 Associations of APOE genotype with amyloid and tau neuropathologies. a** Association of APOE genotype and diffuse amyloid plaque distribution. Within each APOE genotype column, colored rows represent the relative proportion of cases in each severity stage. These proportions are expressed as percentages and add to 100. There are increasing proportions of the most severe pathology (plaque stage 5) from the e2 to e4/e4 genotype groups in stepwise fashion. Amyloid stage 0 = no pathology; stage 5 = severe. **b** Association of APOE genotype and Braak stage. Within each APOE genotype column, colored rows represent the relative proportion of cases in each severity stage. These proportions are expressed as percentages and add to 100. There are increasing proportions of the most severe pathology (Braak stages v and vi) from the e2 to e4/e4 genotype groups in stepwise fashion. Braak stage 0 = no pathology; stage vi = severe neocortical pathology. **c** Association of APOE genotype and neuritic amyloid plaque density level. Within each APOE genotype column, colored rows represent the relative proportion of cases in each severity stage. These proportions are expressed as percentages and add to 100. There are increasing proportions of the most severe pathology (plaque stages v and vi) from the e2 to e4/e4 genotype groups in stepwise fashion. Neuritic plaque stage 0 = no pathology; stage 3 = severe pathology.

**Table 1 APOE logistic regression analysis for AD neuropathologies.**

| Variable | Contrast | OR | CI | Chi-square | p |
|---|---|---|---|---|---|
| APOE genotype contrasts and Thal diffuse plaque | | | | | |
| THAL | E2 v e3 | 0.43 | 0.31–0.60 | 22.38 | <0.0001 |
| THAL | E2 v e4 | 0.11 | 0.10–0.20 | 123.45 | <0.0001 |
| APOE genotype contrasts and Braak stage | | | | | |
| BRAAK | E2 v e3 | 0.54 | 0.39–0.75 | 13.13 | 0.0003 |
| BRAAK | E2 v e4 | 0.12 | 0.11–0.22 | 96.62 | <0.0001 |
| APOE genotype contrasts and neuritic plaque | | | | | |
| NEUR | E2 v e3 | 0.55 | 0.39–0.77 | 11.20 | 0.0008 |
| NEUR | E2 v e4 | 0.14 | 0.10–0.21 | 99.46 | <0.0001 |

e3/e3 genotype in a series of Chi-square analyses and ordinal regressions. For both amyloid pathologies and tau Braak stage, we observed a significant genotype-severity association: a higher proportion of e2/e4 cases had more severe pathologies than did e2/e3 or e3/e3 cases, but this proportion was similar to that found for e3/e4 cases (all Chi square ps < 0.0001 as in Supplementary Tables 5–7). In ordinal regressions, contrasts between e2/e3 and e2/e4 were significant with the latter demonstrating an increased risk of pathology (all ORs > 4.90, all ps < 0.0001) for all three pathologies (Thal diffuse amyloid plaque, Braak stage, neuritic plaques). A similar finding was present for the e3/e3 contrasts though somewhat reduced (ORs > 2.80). Notably, all contrasts between e2/e4 and e3/e4 were nonsignificant, suggesting that both these genotypes conferred more or less equivalent risk for increases in pathology. OR results are in Table 2. Chi square frequency raw data are shown in Supplementary Tables 5–7. The e2/e4 genotype is quite similar to the e3/e4 genotype in being comprised of a high proportion of stages 5 and 6 Braak stages i.e., severe and widespread neocortical involvement (Fig. 3).

**Alpha-synuclein.** For alpha-synuclein inclusion pathology we found a significant difference in frequencies of genotypic association with Lewy body presence and distribution as in

Supplementary Table 8 ($X^2 = 74.69$, df = 9, p < 0.0001). Notably, e4 carriers had higher frequencies of alpha-synuclein pathology when it extended beyond the midbrain to limbic or cortical regions and only a very small proportion (1%) of e4 cases had pathology restricted to the midbrain. This is illustrated in Fig. 4. In ordinal regressions (all adjusted for age at death and sex) we did not find that e2 differed from e3 in reduction of pathology. However, when e4 was contrasted with e2, e2 demonstrated a lower OR of increased extension of pathology (OR = 0.58 CI 0.40–0.86, p = 0.004). A similar pattern was present for the e4 v e3 contrast (data not shown). Thus, e4 demonstrated significantly greater extensions of pathology than either e2 or e3. Because of the substantial co-morbidity between Lewy body pathology and AD, we also adjusted for AD ABC neuropathological change score in a more refined ordinal regression. We found that e2 continued to be associated with significantly reduced pathology when contrasted with e4 (OR = 0.65, CI 0.44–0.97, p = 0.04) and as such e4 promoted pathology (OR = 1.37), independent of AD pathology level compared to e2 and also e3 (data not shown). Age and sex point estimates are in Supplementary Table 9.

**FTLD/tauopathies.** APOE was associated with severity at the trend level for the following FTLD related pathologies: Pick's disease, PSP, and TDP-43. In each case e2 was associated with greater pathology, i.e., was a risk allele. Conversely, e4 genotype cases consistently had the lowest levels of pathology and could thus be viewed as protective. We show these trends in Chi square frequency in Table 3 and Supplementary Tables 10–12. However, and critically, it remained possible that AD pathology confounded this association. Therefore we conducted logistic regressions which were adjusted for ABC neuropathological change scores (as well as sex and age at death). For Picks, PSP, and TDP-43, neuropathological change had significant point estimates, while e2 contrasts were nonsignificant. We show these fully adjusted results in Table 4.

In other Chi-square analyses in Table 3, APOE was not significantly associated with the following FTLD related pathologies: CBD and argyrophilic grain disease.

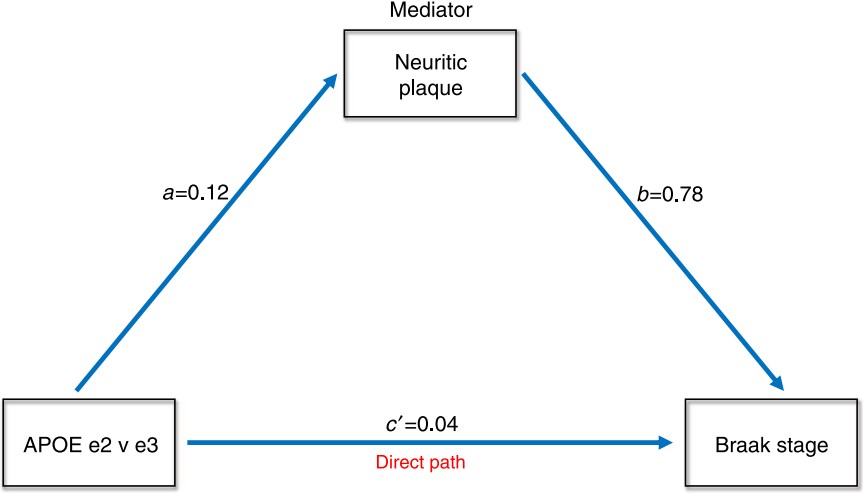

**Fig. 2 Mediation analysis of APOE e2 versus e3 showing both direct and indirect effects (via neuritic plaques) on Braak stage.** Values represent beta weights after adjustment.

**Table 2 APOE e2/e4 logistic regressions for AD neuropathologies.**

| Variable | Contrast | OR | CI | Chi-square | p |
|---|---|---|---|---|---|
| APOE e2/e4 contrasts and neuritic plaque | | | | | |
| NEUR | e24 v e2 | 4.91 | 2.43–9.73 | 24.80 | <0.0001 |
| NEUR | e24 v e3 | 2.85 | 1.56–5.20 | 13.91 | 0.0006 |
| NEUR | e24 v e4 | 1.07 | 0.58–1.97 | 0.10 | 0.84 |
| APOE e2/e4 contrasts and Braak stage | | | | | |
| BRAAK | e2/e4 v e2 | 5.64 | 2.92–10.90 | 30.00 | <0.0001 |
| BRAAK | e2/e4 v e3 | 2.83 | 1.62–4.99 | 15.57 | 0.0003 |
| BRAAK | e2/e4 v e4 | 1.11 | 0.63–1.95 | 0.14 | 0.72 |
| APOE e2/e4 and Thal diffuse plaque | | | | | |
| THAL | e24 v e2 | 5.53 | 2.86–10.73 | 27.93 | <0.0001 |
| THAL | e24 v e3 | 2.41 | 1.37–4.25 | 10.84 | 0.001 |
| THAL | e24 v e4 | 1.12 | 0.60–1.98 | 0.19 | 0.66 |

## Discussion

Several findings derive from our analytic approach and we discuss them in turn. First, e2 was robustly and significantly associated with reductions in AD neuropathology. The frequency of e2 was increased in lower (i.e., regionally more restricted) Braak stages, including stage 0 (no tangle pathology in cortical or MTL regions) when contrasted with both e3 and e4. Similarly, e2 was significantly associated with less extensive plaque pathology (by Thal rating) and lower densities of neuritic plaques than was e3 and e4. The findings using neuropathological data confirm the clinical reports that individuals carrying the APOE e2 allele have a markedly decreased risk of having AD and are consistent with a study showing that e2 was protective against cognitive decline in the NACC clinical database[15].

Mediation analysis indicated that e2 not only has an indirect pathway for reducing Braak stage severity though amyloid, but a highly significant direct path accounting for nearly 60% of the variance. This suggests that diminution in Braak stage in individuals with e2 was not solely due to a reduction in amyloid plaque burden. We have suggested elsewhere as based on e2's post mortem transcriptional upregulation in extra-cellular matrix-related genes that it may play a direct role in modulating tau propagation[7].

We also examined the special case of the rare e2/e4 genotype with respect to its association with AD pathologies. Our results were remarkable and consistent. In three analyses we found that e2/e4 was associated with greater degrees of ABC pathology in contrast to e2/e3 and e3/3 genotypes, and could not be distinguished from e3/e4 cases. Copy number was controlled, increasing the rigor of the approach. These results indicate that the e4 isoform's effects were neither blunted nor otherwise modified by e2 within the same brain, at least insofar as levels of the isoforms were in physiological range. These significant results are similar to those reported by Oveisgharan et al.[16] for amyloid and we now extend them to tau Braak stage.

In an earlier study Serrano-Pozo et al.[17] examined an earlier considerably smaller version ($N = 792$) of the NACC neuropathology database. They found that e2 was significantly associated with reductions in Braak stage, but not not neuritic plaque severity. Consistent with our results they also found significant direct and indirect effects of e2 on Braak stage in mediation analyses. They did not examine e2/e4 genotype, nor Thal plaque scores. In a very recent study Reiman et al.[18] examined over 5000 neuropathologically cases (that included the NACC series); e2/e2 and e2/e3 cases were found to have significant protective on neuritic plaque burden and Braak stage, while the e2/e4 genotype was associated with increased risk, consistent with our study. FTLD/tauopathies were not examined. Based on these results, the aforementioned Oveishagen study, and our own findings, we posit that e2 homozygotes and e2/e3 heterozygote genotypes are associated with greatly diminished odds for AD histopathology, while the e2/e4 genotype is associated with increased odds of pathology, when contrasted with e3 homozygote genotypes. In contrast, e2 is not protective against multiple other proteinopathies, including Lewy body disease and FTLD/tauopathies, suggesting very sharp limits to its advantage.

Fourth, we found robust effects of e4 on the presence of alpha-synuclein pathology in limbic and neocortical regions. Indeed, while 55% of e4 cases exhibited such a pattern, only 1% had a-synuclein restricted to the mid brain alone. Thus, the results are broadly consistent with clinical reports that APOE may be associated with Lewy body dementia[19,20]. E2 and e3 carriers had similar and reduced frequencies of Lewy body pathology contrasted with e4.

Fifth, we examined APOE effects on a variety of FTLD related pathologies including FTLD-tau 3R Pick's, 4R PSP, CBD, and argyrophilic grain disease, and FTLD-TDP-43. No analysis met the study-wide Bonferroni corrected p value, but in analyses of

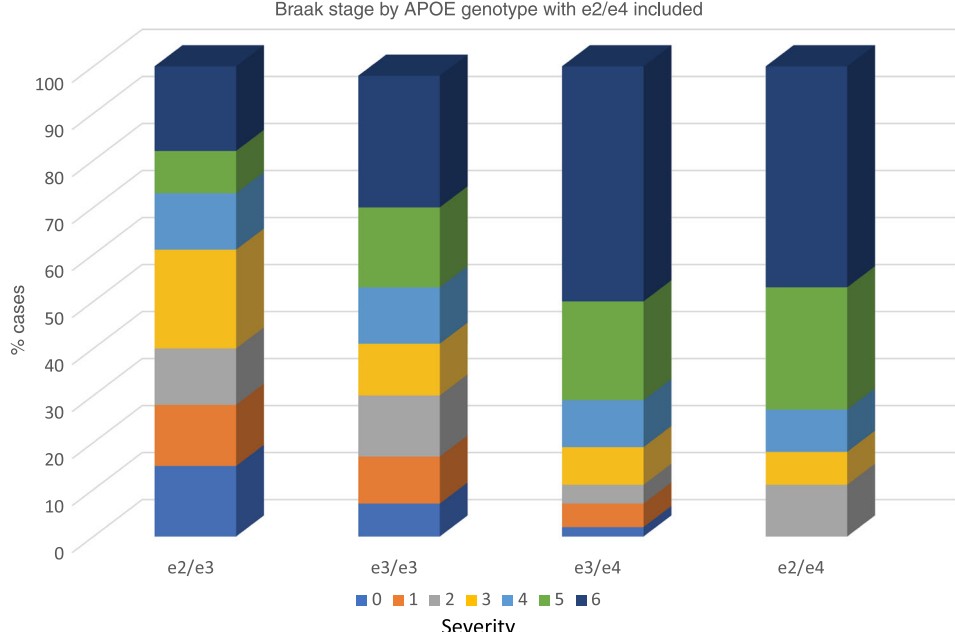

**Fig. 3 Association of APOE e2/e4 genotype with Braak stage.** Within each APOE genotype column, colored rows represent the relative proportion of cases in each severity stage. These proportions are expressed as percentages and add to 100. Note the similarity of e2/e4 cases to e3/e4 cases with respect to high levels of tau pathology. Braak stage 0 = no pathology; stage 6 = severe neocortical pathology.

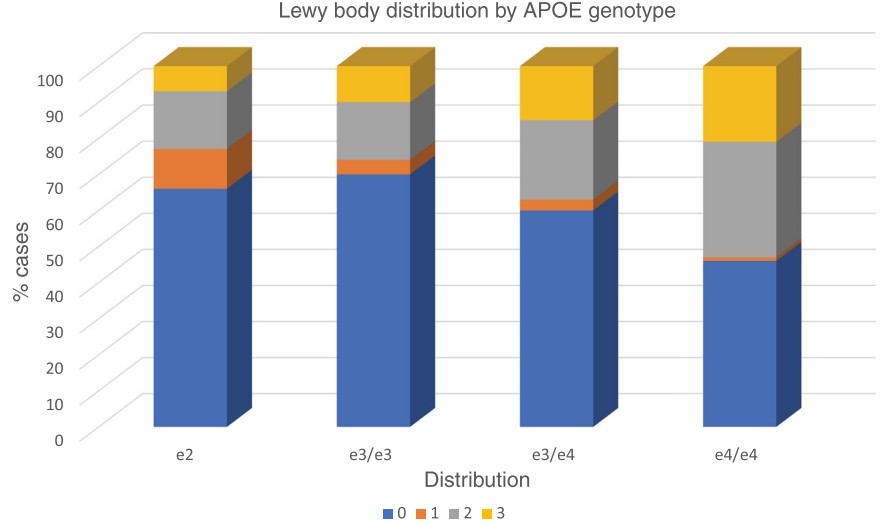

**Fig. 4 Association of APOE genotype with Lewy body pathology.** Within each APOE genotype column, colored rows represent the relative proportion of cases in each severity stage. These proportions are expressed as percentages and add to 100. Note the high proportion of e4 cases with limbic and neocortical pathology and the paucity of e4 cases with pathology restricted to the midbrain. Lewy body stage 0 = no pathology; stage 1 = midbrain; stage 2 = limbic; stage 3 = neocortical.

PSP, Pick's, and TDP-43 pathologies, Chi squares were trend level significant at $p < 0.01$. However, fully adjusted models that included the AD neuropathological change severity scores yielded nonsignificant e2 associations. Thus, the differences between Chi-square analyses and AD adjusted logistic regressions may be the result of 1. complex interactions between FTLD and AD pathologies; 2. an AD ascertainment biases in the sample; or 3. an artifact of statistical adjustment of AD pathology that in conjunction with established e2 effects on AD pathology introduced a confound in the results. Because we could not adjudicate between these we took a more conservative interpretation and considered the results as negative. Nevertheless, the results remain

informative because they demonstrate sharp limits to APOE e2 neuroprotection. Thus, e2 did not offer neuroprotection, even against tau aggregates in Picks and PSP. The literature itself is ambiguous on APOE associations. TDP-43 results in particular are further complicated by its emergence as an age associated pathology, its known relationship to C9orf72 and GRN mutations, and an association with e4[21,22]. Several studies of FTLD and tauopathies, including two in which FTLD was pathologically confirmed, have found e2 to increase risk in pathologically confirmed PSP and CBD[23]; and in a meta-analysis of pathologically confirmed FTLD cases[24]. Chio et al.[25] also found e2 to promote risk of FTLD/ALS (2016). However, other studies have found e4,

**Table 3 FTLD/tauopathy results by Chi square for APOE genotype and pathology.**

| Variable | Stat. | D.F. | Value | Prob. |
|---|---|---|---|---|
| TDP-43 | Chi-square | 3 | 12.7555 | 0.005 |
| Argyrophilic grain | Chi-square | 3 | 4.6548 | 0.19 |
| PSP | Chi-square | 3 | 11.9366 | 0.008 |
| CBD | Chi-square | 3 | 7.5223 | 0.06 |
| Pick's | Chi-square | 3 | 10.5530 | 0.01 |

**Table 4 Odds ratios for APOE e2 in fully adjusted models that included age at death, sex, and ad abc neuropathological change scores (shown as ADNPCS).**

| | OR | CI | p |
|---|---|---|---|
| Pick's[a] | | | |
| e2 v e3 | 0.86 | 0.27–2.77 | 0.80 |
| ADNPCS | 2.97 | 1.58–4.96 | 0.0004 |
| e2 v e4 | 0.51 | 0.06–4.05 | 0.52 |
| ADNPCS | 8.22 | 1.71–39.49 | 0.009 |
| TDP-43[a] | | | |
| e2 v e3 | 1.40 | 0.73–2.71 | 0.31 |
| ADNPCS | 2.31 | 1.76–3.05 | <0001 |
| e2 v e4 | 0.97 | 0.44–2.16 | 0.94 |
| ADNPCS | 2.33 | 1.72–3.17 | <0001 |
| PSP[a] | | | |
| e2 v e3 | 1.43 | 0.59–3.44 | 0.43 |
| ADNPCS | 2.08 | 1.41–3.06 | 0.0002 |
| e2 v e4 | 1.41 | 0.46–4.36 | 0.55 |
| ADNPCS | 2.01 | 1.29–3.15 | 0.002 |

[a]Age but not sex entered the model significantly and Pick's (79.7 years v 69.6 years), TDP-43 (80.3 years v 71.4 years), and PSP (79.8 v. 75.5) cases were younger.

**Table 5 Demographic of the NACC v 10 sample.**

| Genotype | N | Age | Sex (% Male) | AD % |
|---|---|---|---|---|
| e2/e2 & e2/e3 | 130 | 80.3 ± 12.6 | 52 | 40 |
| e3/e3 | 753 | 81.6 ± 11.5 | 55 | 61 |
| e3/e4 | 535 | 79.7 ± 10.1 | 53 | 75 |
| e4/e4 | 139 | 76.3 ± 8.8 | 58 | 96 |

presentation[28,29]. We demonstrated that e2 has robust effects in reducing the probability of an increase in severity in both amyloid and tau staging by about 40–50% when contrasted with e3/e3, and 90% when contrasted with e4. E2 demonstrated both a protective indirect influence on Braak stage with amyloid as a mediator and a substantial direct protective effect. These findings only partially support the amyloid cascade hypothesis which posits that tau aggregation is the result of earlier amyloid mis-processing. In contrast to its generally protective effects, e2 in the presence of e4 in individuals with the e2/e4 genotype was not protective, but rather it "behaved" like the larger e4 group. The protective effects may involve different neurobiological molecular pathways that are not simply an inverse of e4 molecular patho-genesis, consistent with human post mortem transcriptional profiling[7]. We also found strong evidence that e4 may promote the spread of alpha-synuclein pathology outside the midbrain. Finally, e2 was not protective against various FTLD-linked pathologies including 3R and 4R tau forms and TDP-43 pathol-ogies in models adjusting for AD pathology. Moreover, given that Pick's and PSP are generally characterized by predominantly 3R and 4R tau species respectively, and in AD, neurofibrillary tangles are a mixture of 3R and 4R species, it is perhaps unexpected that e2 had no protective effect. These results suggest that there are very clear limits to e2 neuroprotection, including proteinopathies other than AD (alpha-synuclein and tauopathies). Thus, our findings do not fully support the view that shared molecular features among protein aggregation disorders may make possible a unitary approach to treatment of these debilitating disorders.

## Methods

We accessed the NACC Neuropathology Data Set version 10 (December 2016) to conduct this study. It is the most recent version of the NACC neuropathology database with increased granularity for FTLD, tauopathies, and LBD, as well as utilizing ABC (Thal Amyloid, Braak, CERAD Neuritic Plaque) pathological criteria for AD. Autopsies were conducted locally using the NACC Coding Guidebook (January 2014) protocol for uniform collection and ratings of neuropathological data from 39 AD Center sites[30–32]. Semiquantitative ratings were made by immunohistochemistry, histochemistry, microscopic visualization, or visual inspection and appropriate regional examinations. Resulting data were compiled at the central coordinating Center into the publicly available v 10 database.

The sample includes 1557 cases and the data obtained from the updated 2014 NACC neuropathology forms and coding guidebook[30–32]. Severity ratings for each pathology are described in detail in the Codebook. 0 indicates absence of pathology and higher numbers more severe pathology. Severity stages are described in more detail in the Figures.

Inter-site agreement for the ABC pathological criteria for AD were high (Kappa = 0.88) and individual Alzheimer's disease "A,B, and C" scores had agreement kappas ranging from 0.70 to 0.84[33]. Approximately 44% of the sample were e4 carriers and 8% were e2/e2 or e2/e3 carriers. E3 homozygotes comprised 48% of the sample. Approximately 51% of the sample had high probabilities of AD neuro-pathological change based on the ABC score[34,35]. Table 5 lists the APOE genotypic Ns, demographics, and the proportion of cases for each genotype that meet ABC criteria for "high" AD neuropathology change score. We utilized all brains in the v. 10 collection of NACC irrespective of clinical diagnosis or neuropathological diagnosis.

Because of the small number of e2/e2 cases we combined them with e2/e3 cases to form an e2 group. We analyzed e2/e4 cases separately because the risk effects of e4 are opposite to that of e2. Since 95% of the sample was self-identified as Cau-casian, generalizability to other ethnic groups may be limited.

**Statistical approach**. Our statistical plan follows. All analyses were conducted in SAS 9.4.

but not e2, to be a risk allele for FTLD, including a large and recent clinical case control study[26] and a meta analysis of clini-cally defined cases[27].

The current study has several strengths. It is perhaps the largest single study of e2 and e4 effects on multiple post mortem human neuropathologies. It is independent of clinical diagnosis; thus we did not impose any filter or a priori bias on our analyses. Finally, NACC data were collected prospectively and without regard for the specific hypotheses that we tested. Our analyses were rigorous as we used a study-wide Bonferroni correction and all ordinal and logistic regressions adjusted for age of death and sex so that APOE's contribution to pathology could be examined indepen-dent of these factors. However, the case series has an ascertain-ment bias in that all cases came from Alzheimer's Centers and, not surprisingly, ~51% met pathological criteria for high AD neuropathological change scores by ABC system. However, that would not necessarily preclude other pathologies of interest and might even promote them on the basis of being intrinsic to AD or related to protein aggregation. A second limitation is that neu-ropathological assessment may have differed among sites, but the standardized NACC protocol for neuropathological assessment followed at all sites would have decreased such variability.

This study treats de novo the impact of APOE genotype on neurodegenerative pathologies with no a priori biases or filters. Moreover, we believe that this approach not only is important conceptually but also increases statistical power in that the whole cohort is utilized. Such an approach has been utilized to identify the relative contributions of multiple pathologies on age related cognitive decline and dementia without regard for clinical

(1) We first conducted a series of Chi square analyses in order to determine if there were disproportionate frequencies of one or another APOE genotype namely "e2" (comprised of e2/e2s and e2/e3 cases), "e3" (e3 homozygotes), "e3/e4" cases, and "e4/e4" cases associated with neuropathological staging or presence/absence of pathology. The e2/e4 genotype ($N = 46$ cases) was examined in a separate series of analyses.

(2) If findings were positive, we refined our analysis by conducting two planned contrasts in regression models in which e2 was contrasted with e3 and e2 was contrasted with e4 as predictors. In these regressions we adjusted for age at death and sex. If the outcome measure was binary, we utilized logistic regression. If the outcome was ordinal, we utilized ordinal regression.

Given the number of Chi-square analyses that we conducted (12) we used a Bonferroni correction to reduce the probability of type I error. Thus, for 12 Chi square analyses, we set significance at $p < 0.004$. We considered $0.004 < p < 0.01$ trend level significance. For the two planned contrasts using ORs (e2 v e3 and e2 v e4), we considered $p < 0.01$ as significant. $p$ values for ORs were derived from maximum likelihood estimate Wald Chi squares.

We elected to examine APOE genotype associations with the following classes of neuropathologies.

(1) AD-related pathologies based on the robust association of e2 with reduced risk of clinically diagnosed AD and e4 with increased risk for AD. Histopathologies were as defined in the Montine ABC criteria for severity. (A) Diffuse amyloid plaque (Thal stage) is a measure of spread of plaque and higher scores indicate greater spread of pathology. (B) Braak stage is a measure of progression of NFTs and higher level stages indicate spread of pathology to neocortex. (C) Neuritic plaques are a hallmark feature of AD and may have more specificity to AD than diffuse plaques, with higher levels indicating greater density of pathology.

We also examined the impact of APOE on NFTs rated by Braak stage severity in mediation analyses in which amyloid neuritic plaque served as the mediator in an indirect path, based on consistent evidence that amyloid plaques develop prior to NFTs in AD.

Thus, if APOE genotype effects on Braak stage were significant, we sought to determine if there was a significant mediation effect (i.e., an indirect effect) between APOE genotype (e2 v e3) and Braak stage using the Sobel statistic, which is optimal for identifying mediation effects in large samples, while also examining the direct effect of APOE genotype on Braak stage.

For the e2/e4 genotype, analyzed separately, we sought to determine if e2 might in some way minimize e4 related AD pathology. This genotype thus includes both the protective and risk variant isoforms. We sought to determine if the protective variant can to some degree moderate the effects of e4, or if e4 is toxic and can promote pathology independent of e2.

(2) Lewy body disease due to alpha-synuclein aggregations, as it has recently been proposed that there is increased co-morbidity between AD and Lewy body disease[19]. Insofar as APOE e4 is a driver of AD, it may be predicted that it will be associated with LB dementia. Ratings were based on midbrain only, limbic, and neocortical involvement.

(3) FTLD related protein aggregation pathologies including 3R tau Picks disease, other 4R tauopathies and TDP-43 pathology, given suggestions that either AD pathology may promote other protein aggregation neurodegenerative disorders or that protein aggregation disorders share molecular properties that increase risk of co-morbidity. Hence, if APOE e4 promotes a protein aggregation disorder such as AD, it may also promote other such disorders. Similarly, if e2 reduces risk for a protein aggregation disorder such as AD it may reduce risk for other such disorders. All these pathologies were rated as absent/present.

**Reporting summary**. Further information on research design is available in the Nature Research Reporting Summary linked to this article.

## Data availability

The data that supported this study are publicly available by requesting the NACC Neuropathology Data Set v 10 with instructions at https://www.alz.washington.edu/WEB/landingRequest.html.

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

## Acknowledgements

We thank the National Alzheimer's Coordinating Center and Kathryn Gauthreaux, MS, of NACC. The NACC database is funded by NIA/NIH Grant U01 AG016976. NACC data are contributed by the NIA-funded ADCs: P30 AG019610 (PI Eric Reiman, MD), P30 AG013846 (PI Neil Kowall, MD), P30 AG062428-01 (PI James Leverenz, MD) P50 AG008702 (PI Scott Small, MD), P50 AG025688 (PI Allan Levey, MD, PhD), P50 AG047266 (PI Todd Golde, MD, PhD), P30 AG010133 (PI Andrew Saykin, PsyD), P50 AG005146 (PI Marilyn Albert, PhD), P30 AG062421-01 (PI Bradley Hyman, MD, PhD), P30 AG062422-01 (PI Ronald Petersen, MD, PhD), P50 AG005138 (PI Mary Sano, PhD), P30 AG008051 (PI Thomas Wisniewski, MD), P30 AG013854 (PI Robert Vassar, PhD), P30 AG008017 (PI Jeffrey Kaye, MD), P30 AG010161 (PI David Bennett, MD), P50 AG047366 (PI Victor Henderson, MD, MS), P30 AG010129 (PI Charles DeCarli, MD), P50 AG016573 (PI Frank LaFerla, PhD), P30 AG062429-01(PI James Brewer, MD, PhD), P50 AG023501 (PI Bruce Miller, MD), P30 AG035982 (PI Russell Swerdlow, MD), P30 AG028383 (PI Linda Van Eldik, PhD), P30 AG053760 (PI Henry Paulson, MD, PhD), P30 AG010124 (PI John Trojanowski, MD, PhD), P50 AG005133 (PI Oscar Lopez, MD), P50 AG005142 (PI Helena Chui, MD), P30 AG012300 (PI Roger Rosenberg, MD), P30 AG049638 (PI Suzanne Craft, PhD), P50 AG005136 (PI Thomas Grabowski, MD), P30 AG062715-01 (PI Sanjay Asthana, MD, FRCP), P50 AG005681 (PI John Morris, MD), P50 AG047270 (PI Stephen Strittmatter, MD, PhD).

## Author contributions

T.E.G. conceptualized the study, designed the study, conducted the statistical analyses, designed the figures and tables, and wrote the initial draft. E.D.H. critically reviewed the paper. D.P.D. contributed to the design of the study and critically reviewed and edited the paper.

## Competing interests

T.E.G. has received royalties from VeraSci for use of the BACS in clinical trials and was supported by NIA grant R01AG051346. E.D.H. has acted as a consultant to Biogen and Ionis. D.P.D. is a consultant for Acadia, Eisai, Avanir, Genentech, Neuronix, and Grifols.
