## [Peer Review File · Nature Communications]

Reviewers' comments:

Reviewer #1 (Remarks to the Author):

The $\epsilon 2$ allele of APOE is a protective genetic factor against clinical AD. However, it remains elusive whether the APOE2-associated protection is mediated by a reduction of AD pathology (i.e. neuritic plaques and neurofibrillary tangles) in $\epsilon 2$ carriers. Given that APOE2 has been associated with increased longevity, it is plausible APOE2 is generally protective against a spectrum of diseases. Thus, it is also of great interest to test whether APOE2 is protective against other types of neuropathologies. The manuscript by Terry et al. took advantage of the most updated NACC database which is based on a well-established (and the largest) clinical AD cohort to answer these two questions. The authors showed that APOE2 carriers ($\epsilon 2/\epsilon 2$ and $\epsilon 2/\epsilon 3$) have decreased AD pathology but tend to have increased pathology for TDP-43, Pick's bodies and PSP (tau-pathology) in FTLD cases. Interestingly, the protective effect of APOE2 against AD pathology is cancelled out by the presence of $\epsilon 4$ allele (i.e. in $\epsilon 2/\epsilon 4$ individuals).

The research topic of this manuscript is of great interest to the field, however, the study of APOE2 effect on AD pathology utilizing NACC data has been published before with a more detailed analysis showing the similar results reported in this manuscript¹. As such, the novelty of this work is limited. Regardless, there are a few major concerns that need to be addressed by the authors:

1. For results derived from the ordinal/logistic regression analysis in this manuscript (including, AD Histopathology part, section A, B, C; $\epsilon 2/\epsilon 4$ Genotype part; Alpha Synuclein part; FTLD/Tauopathies). The authors need to
 - a. Describe the effect of sex and age at death for each outcome;
 - b. There are multiple descriptions like "with a 57% reduction in the probability of meeting the criteria for any given stage compared to $\epsilon 3$ " in the main text. The description is not correct. Please specify the description as "with a 57% reduction in the probability of meeting the criteria for stage5 compared to $\epsilon 3$ ";
 - c. The authors need to show line plots of the probability of falling into specific category (e.g. severity of diffuse amyloid plaque density) against age at death by the APOE genotype group. The results of males and females should be plotted separately if sex effect is detected.
2. For the mediation analysis, please
 - a. Provide a schematic illustration of the result;
 - b. Include age at death into the model to show the age effect;
 - c. Include sex into the model to show the sex effect.
3. For FTLD/Tauopathies part
 - a. The analyses were based on different sub-cohorts from that used for the AD pathology analysis. The authors need to provide tables to summarize the demographics of the individuals for each pathological cohort;
 - b. APOE2 was shown to increase the risk of different subtypes of FTLD pathologies by the authors. However, the authors did not address whether APOE2 increases the risk of FTLD as a whole. The authors need to compare the APOE2 allele or $\epsilon 2/\epsilon 2 + \epsilon 2/\epsilon 3$ genotype frequency in FTLD population with that in the whole NACC cohort;
 - c. In NACC data, some FTLD cases also have AD pathologies. Including these cases for analysis may confound the effect of APOE2 on FTLD pathology. The authors need to address this problem by excluding such cases.

There are also a few minor concerns in this manuscript:

1. The first sentence of the abstract "Exon 4 of Apolipoprotein E (APOE) gene contains both..." is inaccurate and may be misleading. Exon 4 does not contain two variants of APOE. The two SNPs corresponding to the two variants are located on exon 4.
2. Line 8 in the abstract: " $\epsilon 2$ had large and highly significant protective effects...". The description is not accurate, as only OR which reflects the effect size was given, but the p value/95% confidential interval which reflects the significance was not given.

3. Line 11 in the introduction, reference index "6" should be superscript.
4. Line 2 in results --- AD Histopathology part: "See Table 2A". Table 2A did not show the frequency information described in the first sentence
5. AD Histopathology part, section B: "ORs are in Table 3B." ORs are in Table 2B. Please describe the results of e2 versus e3 in the main text.
6. AD Histopathology part, section C: "ORs are in Table 3C". ORs are in Table 2C.
7. Figure 1C is not described in the main text

Ref. Serrano-Pozo, A., Qian, J., Monsell, S. E., Betensky, R. A. & Hyman, B. T. APOEepsilon2 is associated with milder clinical and pathological Alzheimer disease. *Ann Neurol* 77, 917-929 (2015).

Reviewer #2 (Remarks to the Author):

This is a timely important study to examine the association of APOE2 with multiple neurodegenerative pathologies, leveraging the NACC v. 10 database of 1557 brains that includes 130 e2/e2 or e2/e3 carriers and 679 e4 carriers in order to examine potential neuroprotective effects in multiple proteinopathies irrespective of clinical diagnosis. The authors showed clearly that, for AD-related pathologies of amyloid plaques and neurofibrillary tangle Braak stage, APOE2 had large and highly significant protective effects contrasted with APOE3/3 and APOE4 carriers with odds ratios of about .50 for APOE3 contrasts and .10 for APOE4 contrasts. Strikingly, when they examined APOE2/APOE4 carriers, risk for AD pathologies was similar to that of APOE4 carriers, not APOE2 carriers, suggesting that the APOE4 isoform was dominantly "toxic". APOE4 increased the risk for spread of Lewy bodies to limbic and neocortical regions. However, for FTLN pathologies, APOE2 was associated with increased pathology for TDP-43, Pick's bodies, and progressive supranuclear palsy at trend levels. Based on these observations, the authors concluded that APOE2 was associated with large protective effects on AD neuropathologies, but not on other proteinopathies.

Overall, the study was well designed and conducted, and the data strongly support the main conclusions. This reviewer has no significant comments.

Reviewer #3 (Remarks to the Author):

The authors studied the association between the APOE e2 allele and several neurodegenerative pathologies using the NACC v. 10 database of 1557 brains (130 e2/e2 or e2/e3 carriers and 679 e4 carriers). Their aim was to assess possible neuroprotective effects in multiple proteinopathies regardless of clinical diagnosis. For AD-related pathologies, e2 had highly significant protective effects compared with e3/e3 and e4 carriers (odds ratios of about .50 for e3 contrasts and .10 for e4 contrasts). For e2/e4 carriers, risk for AD pathologies was similar to that of e4 carriers, pointing to the toxicity of the e4 allele. APOE e4 increased the risk for spread of Lewy bodies to limbic and neocortical regions. For fronto-temporal pathologies, e2 was associated with increased pathology for TDP-43, Pick's bodies, and PSP at trend levels. The authors conclude that e2 is associated with large protective effects on AD neuropathologies, but not on other proteinopathies.

This is a clear and important report given the large sample that includes actual brain pathology as well as clinical diagnoses. It thus sheds further light on our understanding of the interaction of the APOE gene on brain pathologies. My only suggestion is to not include information on trends in the abstract.

Reviewer #4 (Remarks to the Author):

In this manuscript, the authors analyzed the impact of the APOE e2 (and in some cases APOE e4) on neuropathological features of Alzheimer's disease (AD) and other neuropathologies in more than 1,500 brain donors. First, they found that donors in the combined APOE e2/e3 and e2/e2 group were distinguished from other APOE genotypes by less extensive amyloid plaques and neurofibrillary tangles, including both plaque-mediated indirect and direct effects on tangle pathology. Second, they found no significant differences in AD pathology between those with the APOE e2/e4 and e3/e4 genotypes, which they suggest supports a potentially toxic gain-of-function effect of APOE e4. Third, they found that APOE e2 is associated with non-significant trends for a paradoxically increased risk of FTLD-related pathologies, including Picks and PSP (primary tauopathies) and TDP-43 pathology. They suggest that the APOE e2 allele's protective effects on AD pathologies and possibly harmful effects on certain other pathologies do not support the idea that the same molecular mechanism are involved in the development and potential treatment of protein aggregation disorders.

This is an interesting report by a thoughtful and productive group. It capitalizes on their novel decision to analyze data from a more inclusive brain donor group, independent of a brain donor's clinical status provide complementary information to and converging support for findings in those other publication. Their analysis includes a broader group of brain donors than in other studies. They also provide some novel information related to the differential impact of APOE2, 3 and 4 on some pathologies (e.g., diffuse plaques and spatial extent of Lewy bodies).

The manuscript has several limitations, some of which could be at least partly addressed: 1) The authors may not have been aware of and do not cite two studies that generated similar findings related to AD pathology, below, which include many but not all of the same brain donors. 2) The failure to detect significant differences between APOE e2/e4 and APOE 3/4 could reflect insufficient statistical power (as suggested below) and, either way, may not be the most compelling argument on behalf of loss-of-function versus gain-of-function APOE e4 effects. 3) The interesting associations between APOE e2 and FTLD-related primary tau and TDP43 pathologies are non-significant trends, and while they are consistent with findings from a recent Nat Commun report, it is again unclear how much overlap there may be in the brain donor groups.

In a pre-print recently posted in MedRxiv, Reiman et al recently reported the effects of APOE e2 and APOE e4 allelic doses on neuritic amyloid plaque and neurofibrillary tangle pathology in over 5,000 brain donors, including cases with the clinical and neuropathological diagnosis of AD dementia and cognitively unimpaired controls without AD and a significant number of donors from the same NACC cohort. Like this study, they showed similar effects of APOE e2 and e4 on amyloid plaque burden, direct and indirect effects on neurofibrillary tangle burden and an effect of APOE e4 on (in this case mostly secondary) Lewy body pathology. In contrast to this manuscript this study, they found that APOE e2/e3 was associated with significantly lower AD dementia risk than APOE e2/e4, either due to greater statistical power or the inclusion of clinical criteria in the selection of cases and controls. I would encourage the authors to incorporate this information and citation, note those features of the current manuscript are novel (e.g., the analysis of a more inclusive group of brain donors irrespective of their clinical features), note similarities and differences in their findings, and acknowledge the partial overlap in brain donor groups.

In an Ann Neurol 2015 article entitled, "APOE e2 is associated with milder clinical and pathological Alzheimer disease," Serrano-Pozo et al conducted a more formal mediation analysis (i.e., incorporating additional potentially confounding effects in their models) to demonstrate direct and indirect effects of APOE e4 gene dose and an additional effect of the APOE e2 itself in nearly 1,000 symptomatic brain donors in the NACC cohort. Again, I would encourage the authors to incorporate this information citation, note those features of the current manuscript are novel (e.g., the analysis of a more inclusive group of brain donors irrespective of their clinical features), note similarities and differences in their findings, and acknowledge the partial overlap in brain donor

groups

I would encourage the authors to note whether there is any overlap between brain donors in this cohort and those in Zhao et al, Nat Commun 2018 or whether the trends provide independent support for their that paradoxical effect of APOE 32 in primary tauopathies; and I would encourage them note novel elements of the current study (e.g., the analysis of TDP-43 effects.

Thank you for the chance to review this interesting report.

"Association of APOE e2 Genotype with Alzheimer's and Non-Alzheimer's Pathologies"

Reviewer #1 (Remarks to the Author):

The $\epsilon 2$ allele of APOE is a protective genetic factor against clinical AD. However, it remains elusive whether the APOE2-associated protection is mediated by a reduction of AD pathology (i.e., neuritic plaques and neurofibrillary tangles) in $\epsilon 2$ carriers. Given that APOE2 has been associated with increased longevity, it is plausible APOE2 is generally protective against a spectrum of diseases. Thus, it is also of great interest to test whether APOE2 is protective against other types of neuropathologies. The manuscript by Terry et al. took advantage of the most updated NACC database which is based on a well-established (and the largest) clinical AD cohort to answer these two questions. The authors showed that APOE2 carriers ($\epsilon 2/\epsilon 2$ and $\epsilon 2/\epsilon 3$) have decreased AD pathology but tend to have increased pathology for TDP-43, Pick's bodies and PSP (tau-pathology) in FTLD cases. Interestingly, the protective effect of APOE2 against AD pathology is cancelled out by the presence of $\epsilon 4$ allele (i.e. in $\epsilon 2/\epsilon 4$ individuals).

The research topic of this manuscript is of great interest to the field, however, the study of APOE2 effect on AD pathology utilizing NACC data has been published before with a more detailed analysis showing the similar results reported in this manuscript¹. As such, the novelty of this work is limited. Regardless, there are a few major concerns that need to be addressed by the authors:

1. For results derived from the ordinal/logistic regression analysis in this manuscript (including, AD Histopathology part, section A, B, C; $\epsilon 2/\epsilon 4$ Genotype part; Alpha Synuclein part; FTLD/Tauopathies). The authors need to

a. Describe the effect of sex and age at death for each outcome;

In our original submission we noted that we adjusted for age and sex in our ordinal regression models. We now show the odds ratio, and p values for them in a revised Table 2.

b. There are multiple descriptions like "with a 57% reduction in the probability of meeting the criteria for any given stage compared to e3" in the main text. The description is not correct. Please specify the description as "with a 57% reduction in the probability of meeting the criteria for stage5 compared to e3";

In consultation with our biostatistician we have now revised statements throughout. For example, we now state: "... with a 57% reduction in the odds ratio of meeting the criteria for any given stage compared to e3"

c. The authors need to show line plots of the probability of falling into specific category (e.g. severity of diffuse amyloid plaque density) against age at death by the APOE genotype group. The results of males and females should be plotted separately if sex effect is detected.

These plots are now provided as an attachment at the end of this response. Given their length we have not included them in our ms, but could include them as a Supplement if requested to do so. Age at death was consistently unrelated to pathology. In general males had more pathology than females in this sample. Nomenclature on the SAS plots is as follows: apoe= 0 is the e2 genotype group, 1 is the e3/e3 group, 2 is the e3/e4 group, and 3 is the e4/e4 group. For sex,

1=female and 2=male

2. For the mediation analysis, please

a. Provide a schematic illustration of the result;

Now done as the new Figure 4

b. Include age at death into the model to show the age effect;

All paths were adjusted for age at death and sex in the original submission. We apologize for not noting this.

c. Include sex into the model to show the sex effect.

As above.

3. For FTLD/Tauopathies part

a. The analyses were based on different sub-cohorts from that used for the AD pathology analysis. The authors need to provide tables to summarize the demographics of the individuals for each pathological cohort;

We note that for the FTLD/tauopathies total Ns were nearly that of the AD sample (TDP 43 N=1147 negative and 103 positive; PSP N=1199 neg and 51 positive; Picks N=1225 neg and 25 positive) given our transdiagnostic approach. Basic sex ratio differed trivially in the positive groups from the negative groups (all X^2 s<.244, all p values>.12). For age at death, FTLD cases were younger (with TDP PSP and Picks age differences being significant) now noted in the new Table 5. Note that in all our regressions both sex and age were included in the model. Additionally and importantly, please see our response to c.

b. APOE2 was shown to increase the risk of different subtypes of FTLD pathologies by the authors. However, the authors did not address whether APOE2 increases the risk of FTLD as a whole. The authors need to compare the APOE2 allele or $\epsilon 2/\epsilon 2 + \epsilon 2/\epsilon 3$ genotype frequency in FTFD population with that in the whole NACC cohort;

Please see our response to c.

c. In NACC data, some FTLD cases also have AD pathologies. Including these cases for analysis may confound the effect of APOE2 on FTLD pathology. The authors need to address this problem by excluding such cases.

This is a very salient point. We reconducted our logistic regression analyses for Picks, PSP, and TDP 43 by adjusting for the ABC neuropathological change score. After doing so we found that 1. This score accounted for most of the variance; and 2. APOE e2 genotype was no longer a significant risk predictor. We show these results in Table 5. We have tempered our conclusions in the Discussion relating to e2 as a risk factor for FTLD accordingly. We now state in the Discussion:

“However, fully adjusted models that included the AD neuropathological change severity scores yielded non-significant e2 associations. Thus, the differences between chi-square analyses and AD adjusted logistic regressions may be the result of 1. complex interactions between FTLD and AD pathologies; 2. an AD ascertainment biases in the sample; or 3. an artifact of statistical adjustment of AD pathology that in conjunction with established e2 effects on AD pathology introduced a confound in the results. Because we could not adjudicate between these we took a more conservative interpretation and considered the results negative. Nevertheless the results remain informative because they demonstrate sharp limits to APOE e2 neuroprotection. Thus, e2 did not offer neuroprotection even against tau based aggregates in Picks and PSP.”

There are also a few minor concerns in this manuscript:

1. The first sentence of the abstract “Exon 4 of Apolipoprotein E (APOE) gene contains both...” is inaccurate and may be misleading. Exon 4 does not contain two variants of APOE. The two SNPs corresponding to the two variants are located on exon 4.

Now revised

2. Line 8 in the abstract: “e2 had large and highly significant protective effects...”. The description is not accurate, as only OR which reflects the effect size was given, but the p value/95% confidential interval which reflects the significance was not given.

Now changed to: After adjusting for AD neuropathology in logistic regressions e2 associations were no longer significant.

3. Line 11 in the introduction, reference index “6” should be superscript.

Corrected

4. Line 2 in results --- AD Histopathology part: “See Table 2A”. Table 2A did not show the frequency information described in the first sentence

5. AD Histopathology part, section B: “ORs are in Table 3B.” ORs are in Table 2B. Please describe the results of e2 versus e3 in the main text.

6. AD Histopathology part, section C: “ORs are in Table 3C”. ORs are in Table 2C.

7. Figure 1C is not described in the main text

For 4-7. Now corrected. We apologize for these typos.

Ref. Serrano-Pozo, A., Qian, J., Monsell, S. E., Betensky, R. A. & Hyman, B. T. APOEepsilon2 is associated with milder clinical and pathological Alzheimer disease. *Ann Neurol* 77, 917-929 (2015).

Now cited and commented upon in the Discussion. We apologize for the omission of this important paper.

Reviewer #2 (Remarks to the Author):

This is a timely important study to examine the association of APOE2 with multiple neurodegenerative pathologies, leveraging the NACC v. 10 database of 1557 brains that includes 130 e2/e2 or e2/e3 carriers and 679 e4 carriers in order to examine potential neuroprotective effects in multiple proteinopathies irrespective of clinical diagnosis. The authors showed clearly that, for AD-related pathologies of amyloid plaques and neurofibrillary tangle Braak stage, APOE2 had large and highly significant protective effects contrasted with APOE3/3 and APOE4 carriers with odds ratios of about .50 for APOE3 contrasts and .10 for APOE4 contrasts. Strikingly, when they examined APOE2/APOE4 carriers, risk for AD pathologies was similar to that of APOE4 carriers, not APOE2 carriers, suggesting that the APOE4 isoform was dominantly “toxic”. APOE4 increased the risk for spread of Lewy bodies to limbic and neocortical regions. However, for FTLN pathologies, APOE2 was associated with increased pathology for TDP-43, Pick’s bodies, and progressive supranuclear palsy at trend levels. Based on these observations, the authors concluded that APOE2 was associated with large protective effects on AD neuropathologies, but not on other proteinopathies.

Overall, the study was well designed and conducted, and the data strongly support the main conclusions. This reviewer has no significant comments.

We appreciate the Reviewer for his/her very positive comments.

Reviewer #3 (Remarks to the Author):

The authors studied the association between the APOE e2 allele and several neurodegenerative pathologies using the NACC v. 10 database of 1557 brains (130 e2/e2 or e2/e3 carriers and 679 e4 carriers). Their aim was to assess possible neuroprotective effects in multiple proteinopathies regardless of clinical diagnosis. For AD-related pathologies, e2 had highly significant protective effects compared with e3/e3 and e4 carriers (odds ratios of about .50 for e3 contrasts and .10 for e4 contrasts). For e2/e4 carriers, risk for AD pathologies was similar to that of e4 carriers, pointing to the toxicity of the e4 allele. APOE e4 increased the risk for spread of Lewy bodies to limbic and neocortical regions. For fronto-temporal pathologies, e2 was associated with increased pathology for TDP-43, Pick's bodies, and PSP at trend levels. The authors conclude that e2 is associated with large protective effects on AD neuropathologies, but not on other proteinopathies.

This is a clear and important report given the large sample that includes actual brain pathology as well as clinical diagnoses. It thus sheds further light on our understanding of the interaction of the APOE gene on brain pathologies. My only suggestion is to not include information on trends in the abstract.

We thank the Reviewer for his/her positive comments. We have now removed trend information from the Abstract.

Reviewer #4 (Remarks to the Author):

In this manuscript, the authors analyzed the impact of the APOE e2 (and in some cases APOE e4) on neuropathological features of Alzheimer's disease (AD) and other neuropathologies in more than 1,500 brain donors. First, they found that donors in the combined APOE e2/e3 and e2/e2 group were distinguished from other APOE genotypes by less extensive amyloid plaques and neurofibrillary tangles, including both plaque-mediated indirect and direct effects on tangle pathology. Second, they found no significant differences in AD pathology between those with the APOE e2/e4 and e3/e4 genotypes, which they suggest supports a potentially toxic gain-of-function effect of APOE e4. Third, they found that APOE e2 is associated with non-significant trends for a paradoxically increased risk of FTLD-related pathologies, including Picks and PSP (primary tauopathies) and TDP-43 pathology. They suggest that the APOE e2 allele's protective effects on AD pathologies and possibly harmful effects on certain other pathologies do not support the idea that the same molecular mechanism are involved in the development and potential treatment of protein aggregation disorders.

This is an interesting report by a thoughtful and productive group. It capitalizes on their novel decision to analyze data from a more inclusive brain donor group, independent of a brain donor's clinical status provide complementary information to and converging support for findings in those other publication. Their analysis includes a broader group of brain donors than in other studies. They also provide some novel information related to the differential impact of APOE2, 3 and 4 on some pathologies (e.g., diffuse plaques and spatial extent of Lewy bodies).

We thank the Reviewer for these comments.

The manuscript has several limitations, some of which could be at least partly addressed: 1) The authors may not have been aware of and do not cite two studies that generated similar findings related to AD pathology, below, which include many but not all of the same brain donors. 2) The failure to detect significant differences between APOE e2/e4 and APOE 3/4 could reflect insufficient statistical power (as suggested below) and, either way, may not be the most compelling argument on behalf of loss-of-function versus gain-of-function APOE e4 effects. 3) The interesting associations between APOE e2 and FTL-related primary tau and TDP43 pathologies are non-significant trends, and while they are consistent with findings from a recent Nat Commun report, it is again unclear how much overlap there may be in the brain donor groups.

We agree with the Reviewers point about gain of function v loss of function and have removed the sentence. In the Discussion we now note possible overlap in the samples used in this study vis a vis the Serrano Pozzo and Reiman studies. We have substantially revised our FTL analyses (see Reviewer 1) and considerably tempered all conclusions, noting rather that there appear to be clear limits to e2 neuroprotection.

In a pre-print recently posted in MedRxiv, Reiman et al recently reported the effects of APOE e2 and APOE e4 allelic doses on neuritic amyloid plaque and neurofibrillary tangle pathology in over 5,000 brain donors, including cases with the clinical and neuropathological diagnosis of AD dementia and cognitively unimpaired controls without AD and a significant number of donors from the same NACC cohort. Like this study, they showed similar effects of APOE e2 and e4 on amyloid plaque burden, direct and indirect effects on neurofibrillary tangle burden and an effect of APOE e4 on (in this case mostly secondary) Lewy body pathology. In contrast to this manuscript this study, they found that APOE e2/e3 was associated with significantly lower AD dementia risk than APOE e2/e4, either due to greater statistical power or the inclusion of clinical criteria in the selection of cases and controls. I would encourage the authors to incorporate this information and citation, note those features of the current manuscript are novel (e.g., the analysis of a more inclusive group of brain donors irrespective of their clinical features), note similarities and differences in their findings, and acknowledge the partial overlap in brain donor groups.

We agree and have now incorporated findings from this excellent paper into our Discussion. As a note here, we actually found similar results in that e2/e3 was protective but e2/e4 was a risk variant (ie, it promoted risk) with similar odds ratios to e3/e4 for AD pathologies.

In an Ann Neurol 2015 article entitled, "APOE e2 is associated with milder clinical and pathological Alzheimer disease," Serrano-Pozo et al conducted a more formal mediation analysis (i.e., incorporating additional potentially confounding effects in their models) to demonstrate direct and indirect effects of APOE e4 gene dose and an additional effect of the APOE e2 itself in nearly 1,000 symptomatic brain donors in the NACC cohort. Again, I would encourage the authors to incorporate this information citation, note those features of the current manuscript are novel (e.g., the analysis of a more inclusive group of brain donors irrespective of their clinical features), note similarities and differences in their findings, and acknowledge the partial overlap in brain donor groups

Now included and discussed. See our response to Reviewer 1. This is an important paper and we apologize for its omission from our original ms.

I would encourage the authors to note whether there is any overlap between brain donors in this cohort and those in Zhao et al, Nat Commun 2018 or whether the trends provide independent support for their that paradoxical effect of APOE 32 in primary tauopathies; and I would encourage them note novel elements of the current study (e.g., the analysis of TDP-43 effects. *As noted in our response to Rev 1, our FTLD results were attenuated and no longer positive in new regressions that adjusted for the ABC neuropathological change score. As best as we could determine, Zhao used a Mayo brain bank. We do not know if Mayo contributed to the NACC neuropathology database.*

Thank you for the chance to review this interesting report.

Age and Sex Scattergrams for APOE Genotypes

REVIEWERS' COMMENTS:

Reviewer #1 (Remarks to the Author):

The authors have adequately addressed my concerns.

Reviewer #2 (Remarks to the Author):

This is a timely important study to examine the association of APOE2 with multiple neurodegenerative pathologies. The revised manuscript is more clear and balanced.

This reviewer has no further comments.

Reviewer #3 (Remarks to the Author):

Thank you for your detailed responses to the reviewer critiques.